# Impact of Heat Stress on Poultry Health and Performances, and Potential Mitigation Strategies

**DOI:** 10.3390/ani10081266

**Published:** 2020-07-24

**Authors:** Sanjeev Wasti, Nirvay Sah, Birendra Mishra

**Affiliations:** Department of Human Nutrition, Food and Animal Sciences, University of Hawaii at Manoa, Honolulu, HI 96822, USA; swasti@hawaii.edu (S.W.); nirvay@hawaii.edu (N.S.)

**Keywords:** heat stress, mitigation, oxidative stress, physiological changes, poultry

## Abstract

**Simple Summary:**

High environmental temperature alters the poultry health and performances by causing heat stress. Heat stress elicits physiological, behavioral, and production changes in poultry. This review article summarizes these changes along with the strategies that have been used in the poultry industry to ameliorate the adverse effects of heat stress in poultry.

**Abstract:**

Heat stress is one of the major environmental stressors in the poultry industry resulting in substantial economic loss. Heat stress causes several physiological changes, such as oxidative stress, acid-base imbalance, and suppressed immunocompetence, which leads to increased mortality and reduced feed efficiency, body weight, feed intake, and egg production, and also affects meat and egg quality. Several strategies, with a variable degree of effectiveness, have been implemented to attenuate heat stress in poultry. Nutritional strategies, such as restricting the feed, wet or dual feeding, adding fat in diets, supplementing vitamins, minerals, osmolytes, and phytochemicals, have been widely studied and found to reduce the deleterious effects of heat stress. Furthermore, the use of naked neck (Na) and frizzle (F) genes in certain breed lines have also gained massive attention in recent times. However, only a few of these strategies have been widely used in the poultry industry. Therefore, developing heat-tolerant breed lines along with proper management and nutritional approach needs to be considered for solving this problem. Thus, this review highlights the scientific evidence regarding the effects of heat stress on poultry health and performances, and potential mitigation strategies against heat stress in broiler chickens and laying hens.

## 1. Introduction

The poultry industry is growing across the world to fulfill the increasing demands of poultry meat and eggs. Poultry meat contains a low amount of saturated fatty acids and is rich in protein, vitamins, and minerals [1]. Similarly, poultry eggs are the most affordable source of animal protein [2]. Besides vitamins, minerals, and proteins, eggs are also rich in antioxidants such as lutein and zeaxanthin, which possess major benefits for eye health [3]. Considering these facts, the global consumption of poultry meat and eggs have doubled in the past decade and is expected to be doubled by 2050 [4]. To fulfill the demands, there has been an immense improvement in chicken genetics in the past decade. Broiler chickens, which weighed around 900 g in 56 days in the 1950s, were around 4202 g in 2005 [5]. Similarly, laying hens in the early 1900s used to lay 150 eggs per year while current commercial laying hens lay around 300 eggs annually [6]. These improved broilers and laying hens have higher metabolic rates and production performances [7]. Due to a higher metabolic rate, they produce more body heat and are prone to heat stress. High stocking density of birds, along with the high ambient temperature, increases the propensity of heat stress [8].

Heat stress is a major problem in the poultry industry affecting the health and performances of poultry. As of 2003, heat stress resulted in annual economic losses of $128 to $165 million in the poultry industry [9], and with the rise of global temperature, this number is speculated to increase in the coming years. Heat stress is a condition where chickens are unable to maintain a balance between body heat production and heat loss. Heat stress results from the interaction of different factors such as high environmental temperature, humidity, radiant heat, and airspeed; among them, high ambient temperature plays a significant role [10]. The normal body temperature of the chicken is around 41–42 °C, and the thermoneutral temperature to maximize growth is between 18–21 °C [11]. Studies have shown that any environmental temperature higher than 25 °C elicits heat stress in poultry [12]. As there is extensive scientific evidence about the detrimental effects of heat stress on poultry health and performances, and potential mitigation strategies, it is crucial to summarize these findings for poultry researchers and industry. Therefore, the objectives of this review paper are to summarize the (1) physiological, neuroendocrine, and behavioral changes in poultry under heat stress, and (2) potential mitigation strategies against heat stress in broiler chickens and laying hens.

## 2. Biological Changes in Poultry Due to Heat Stress

Heat stress in poultry results in several behavioral, physiological, and neuroendocrine changes that influence health and performances (Figure 1).

### 2.1. Physiological Changes

Major physiological changes that take place in the heat-stressed birds are:

#### 2.1.1. Oxidative Stress

Reactive oxygen species (ROS) are free radicals and peroxides produced typically within the cells during regular metabolism. They are essential for many cellular processes such as cytokine transcription, immunomodulation, and ion transportation. The excess ROS produced within cells are eliminated by physiological detoxifying mechanisms present within the cells. During the thermoneutral condition, activation of transcriptional factor Nrf2 causes the additional synthesis of a group of antioxidant molecules, which deals with increased ROS produced inside the cell [13]. However, due to the imbalance between these systems, either by higher production of ROS or by a decrease in the effectiveness of the antioxidant defense system, the cells are exposed to stress conditions commonly known as oxidative stress [14,15]. Previous studies in poultry have shown that heat stress is associated with cellular oxidative stress [13,16]. Excess free radicals produced during oxidative stress damage all the components of the cells including proteins, lipids, and DNA (Figure 2). Effects of oxidative stress depend upon its severity and range from small reversible changes to apoptosis and cell death in the case of severe oxidative stress [17]. The oxidative stress in poultry is associated with biological damage, severe health disorders, lower growth rates, and economic losses [16].

#### 2.1.2. Acid-Base Imbalance

Birds lack sweat glands and have feathers throughout the body [18]. Those features impair thermoregulation, and as a consequence, they need to release heat via active mechanism (i.e., panting) during higher ambient temperature. Panting is a phenomenon exhibited by the birds by opening their beak to increase respiration rate and evaporative cooling from the respiratory tract. During panting, excretion of CO_2_ occurs at a greater rate than the cellular production of CO_2_, which alters the standard bicarbonate buffer system in the blood. The reduction of CO_2_ leads to a decrease in the concentration of carbonic acids (H_2_CO_3_) and hydrogen ions (H^+^). In contrast, the concentration of the bicarbonate ions (HCO_3_^−^) is increased; thus, raising the blood pH, i.e., the blood becomes alkaline. To cope with this situation and maintain the normal blood pH, birds will start excreting more amount of HCO_3_^−^ and retain H^+^ from the kidney. The elevated H^+^ alters the acid-base balance leading to respiratory alkalosis and metabolic acidosis (Figure 3) and is associated with the decline in production performances of poultry [19].

#### 2.1.3. Suppressed Immunocompetence

Heat stress is known to suppress immunity in the chicken [10]. As a result, the prevalence of contagious and infectious poultry diseases, such as Newcastle disease (ND) and Gumboroo disease, is relatively higher during the summer season in tropical countries [20]. Besides this, the size of immune-related organs such as the spleen, thymus, and lymphoid organs are also regressed in the heat-stressed birds [21,22]. The level of antibodies was also lowered in the heat-stressed birds [23]. Likewise, total white blood cell counts (WBC) are significantly lowered, whereas the heterophils to lymphocytes (H/L) ratio is higher in heat-stressed birds [24].

### 2.2. Neuroendocrine Changes

The neuroendocrine system plays a crucial role in maintaining homeostasis and normal physiological functioning of birds during heat stress. In birds, the sympathoadrenal medullary (SAM) axis is activated and regulates homeostasis during the early stage of heat stress. The increase in ambient temperature is perceived by the sympathetic nerves, which transmit the impulse to the adrenal medulla. The adrenal medulla increases the secretion of catecholamines, which cause a surge of glucose release in the blood, deplete liver glycogen, reduce muscle glycogen, increase respiration rate, vasodilate the peripheral blood vessels, and increase neural sensitivity to cope with the stress [11,25]. As stress persists for a more extended period, the hypothalamic-pituitary-adrenal (HPA) axis is activated. In response to the stress, corticotrophin-releasing hormone (CRH) is secreted from the hypothalamus, which triggers the release of an adrenocorticotrophic hormone (ACTH) from the pituitary. ACTH increases the production and release of corticosteroid by the adrenal glands [26]. Corticosteroid stimulates gluconeogenesis to increase plasma glucose levels [11,25]. Thyroid hormones, triiodothyronine (T3) and thyroxine (T4), released by the thyroid gland, also play a critical role in maintaining metabolic rate. Previous studies have shown that T3 concentrations were lowered in the heat-stressed birds [27,28,29], whereas T4 concentrations were found inconsistent in different studies [27]. The reduction of T3 concentration during heat stress is due to a decrease in peripheral deiodination of T4 to T3 [30]. There is also a difference in T3 secretion between selected breeds and native breeds. The plasma T3 levels in dwarfs (dw) chicken are usually less than half of the levels in selected breeds of chicken. Dwarf gene (dw) is found to inhibit the conversion of T4 to T3 in peripheral tissue, resulting in a lower T3 level in dwarfs [31]. Melesse et al. [32] also reported a lower level of T3 in Naked neck laying hens as compared to Lohman white and New Hampshire laying hens. Besides this, the secretion of the gonadotrophin-releasing hormone is also found to be impaired in heat-stressed birds [33]. Moreover, sex hormones such as plasma progesterone, testosterone, and estradiol were also found to be lowered in heat-stressed White Leghorns [34]. These hormonal changes are responsible for reduced growth performance [22] and reproductive efficiency [35] of hyperthermic birds.

### 2.3. Behavioral Changes

When birds are exposed to a higher environmental temperature than their thermoneutral temperature, they try to dissipate excess heat produced inside the body, which is manifested by specific behavioral changes in birds. Chickens in the thermal stress condition spend less time walking and standing, consume less amount of feed and more water, spread wings, and cover their body surface in the litter. Furthermore, the characteristic signs of panting are also observed in heat-stressed birds [10].

These major physiological, neuroendocrine, and behavioral changes lead to increased mortality, decreased feed intake, reduced final body weight, decreased quality of meat and eggs, and increased feed conversion ratio (FCR) in poultry. Thus, heat stress has been of paramount importance in the poultry industry considering global warming and economic losses. To cope with this problem, different strategies have been employed by researchers and farmers [36].

## 3. Potential Strategies to Mitigate Heat Stress in Poultry

Major strategies that have been used to mitigate the detrimental effects of heat stress in poultry are discussed in this review paper.

### 3.1. Feeding Strategies

#### 3.1.1. Feed Restriction

Restricting the feed during the hotter period of the day has been a common practice in poultry production. In this practice, feed intake is reduced by withdrawing feed for a certain period (generally 8 a.m. to 5 p.m.) to reduce the metabolic rate of birds. Feed restriction is found to reduce rectal temperature, minimize mortality [37,38] and decrease abdominal fat [39] in heat-stressed broilers. Uzum et al. [38] found that restricting the availability of feed to 8 h a day during the hot periods in broilers improved feed efficiency and shortened tonic immobility; a measure to determine fearfulness in which birds are placed on its back for observing righting reflex. Similarly, in the case of broiler hens, limiting feed provision was found to reduce heat production by 23% [40]. Yet, this approach is not widely used in the poultry industry, as it results in reduced growth rate and delayed marketing age of the birds [37,38,41,42].

#### 3.1.2. Dual Feeding Regime

Practical observations have shown that feed restriction results in overcrowding and rush at a re-feeding time resulting in some additional mortality. Thus, the dual feeding regime has been devised to ensure birds have access to feed throughout the day. The thermic effects of proteins are higher than carbohydrates and produce higher metabolic heat [43]. Taking this into account, the protein-rich diet is provided during cooler times and the energy-rich diet during the warmer period of the day. Studies have shown that providing a protein-rich diet from 4 p.m. to 9 a.m. and an energy-rich diet during the 9 a.m. to 4 p.m. heat stress period was found to reduce the body temperature [44,45] and mortality in the heat-stressed broilers [44]. However, this approach could not enhance growth and feed efficiency in heat-stressed birds [45].

#### 3.1.3. Wet Feeding

During heat stress, birds lose a high amount of water through the respiratory tract, and there is a marked increase in water intake to restore thermoregulatory balance [18]. Adding water in the feed helps increase water intake and reduces viscosity in the gut resulting in the faster passage of the feed. Wet feeding stimulates pre-digestion, improves absorption of the nutrients from the gut, and accelerates the action of the digestive enzyme on the feed [46]. In broilers, wet feeding improved the feed intake, body weight, and weight of the GI tract [47,48,49]. In laying hens, feeding of wet feed during the high temperature increased dry matter intake, egg weight, and egg production [36]. Although this approach was found to have beneficial effects in heat-stressed birds, it is less common among poultry farmers, as there is a risk of fungal growth in the feed causing mycotoxicosis in the birds.

#### 3.1.4. Adding Fat in the Diet

Higher energy diets were effective in partially mitigating the effects of heat stress in poultry. During metabolism, fat produces lower heat increment as compared to protein and carbohydrates [50]. Considering this fact, supplementation of fat in the diet has been a general practice in the hot climatic regions to increase the energy level and diminish the detrimental effects of heat stress. Supplementation of fat in the poultry diet not only helps to increase the nutrient utilization in the GI tract by lowering the rate of food passage [51] but also helps to increase the energy value of the other feed constituents [52,53]. Adding fat at the level of 5% to the diet in heat-stressed laying hens was found to increase feed intake by 17% [54]. Similarly, significant improvement in the broiler performance was observed when the 5% fat diet was provided [55]. Attia et al. [56] also reported that increasing the oil supplementation in the higher protein concentration diet relieved the negative effects of chronic heat stress on broiler performance, meat lipids, and physiological and immunological traits. In addition to these benefits, adding fat significantly increased abdominal fat in heat-stressed broilers [55].

#### 3.1.5. Supplementation of Vitamins, Minerals, and Electrolytes

##### Vitamin E

Vitamin E (alpha-tocopherol) is a fat-soluble vitamin that has antioxidant activity and helps to scavenge free radicals produced inside the cell. Vitamin E is found to modulate inflammatory signaling, regulate the production of prostaglandins, cytokines, and leukotrienes, and also improve the phagocytic activity of macrophages in broiler chickens [57]. Furthermore, Vitamin E also helps to improve immunity by inducing proliferation of lymphocytes [58,59]. Dietary supplementation of vitamin E in heat-stressed laying hens is found to improve egg production, egg weight, eggshell thickness, egg specific gravity, and Haugh unit [60]. Bollengier-Lee [61] concluded that dietary supplementation of 250 mg vitamin E/kg of feed is optimum for alleviating adverse effects of chronic heat stress in laying hens. The liver is an essential organ for egg formation as it helps in the synthesis and release of egg yolk protein-vitellogenin. Yardibi et al. [62] stated that vitamin E helps to improve the egg production by preventing liver damage in the heat-stressed birds and thus, facilitate the synthesis and release of vitellogenin [63]. Similarly, broilers supplemented with vitamin E (250 mg/kg of feed) have reduced liver and serum malondialdehyde (MDA) concentration, and increased serum and liver vitamin E and A concentration in heat stress conditions [64], as summarized in Table 1. The combination of vitamin E (100 mg/kg of feed), vitamin C (200 mg/kg of feed) and probiotics (*Saccharomyces cerevisiae* and *Lactobacillus acidophilus* at 2 g/kg of feed) was found to be more effective to attenuate negative effects of heat stress in broilers under chronic condition [65].

##### Vitamin A

Vitamin A is associated with antibody production and T cell proliferation [66]. Vitamin A is the most effective antioxidant at low oxygen tensions, which is found to quench singlet oxygen, neutralize thiyl radicals, and combine with and stabilize peroxyl radicals [67]. In a study, supplementation of a higher level of vitamin A (6000 and 9000 IU/kg of feed) was found to increase the egg weight in the heat-stressed laying hens [68]. They also reported that hens exposed to heat stress immediately after NDV (Newcastle disease virus) vaccination require a higher amount of vitamin A for an adequate level of antibody production. In broilers, supplementation of vitamin A (IU/kg of feed) was found to increase the live weight gain, improve feed efficiency, and decrease the serum MDA concentration in the heat-stressed birds [69].

##### Vitamin C 

Vitamin C is a water-soluble antioxidant that protects against oxidative stress by scavenging ROS, neutralizing vitamin E-dependent hydroperoxyl radicals, and protecting proteins from alkylation and by electrophilic lipid peroxidation products [70]. Vitamin C is also known to improve immunity by enhancing the differentiation and proliferation of T and B cells [71]. Although poultry can synthesize vitamin C, the amount is limited during heat stress conditions [72]. Thus, dietary supplementation of vitamin C is an effective strategy to reduce the harmful effects of heat stress in poultry. Supplementation of vitamin C (250 mg/kg of feed) improved growth rate, nutrient utilization, egg production, and quality, immune response, and antioxidant status in heat-stressed birds [72]. Dietary supplementation of vitamin C lowered the serum concentration of MDA, homocysteine, and adrenal corticotropin hormone in heat-stressed Japanese quail [73]. In broilers, dietary supplementation of 200 mg ascorbic acid per kg of feed improved body weight gain and FCR [74].

##### Zinc

Zinc is an essential nutrient required for the enzymatic activity for more than 300 different enzymes. Zinc is associated with the antioxidant defense system, immune function, and skeletal development [75]. Zinc also plays an essential role in the synthesis of metallothionein, which acts as a free radical scavenger [76]. Moreover, zinc is an integral component of carbonic anhydrase, the enzyme that catalyzes the formation of carbonates, an essential compound for eggshell mineralization. [77]. The supplementation of zinc helped to suppress the free radicals by being part of superoxide dismutase, glutathione, glutathione S-transferase, and hemeoxygenase-1 [78]. In broilers, supplementation of the organic form of zinc (40 mg/kg of feed) was effective in improving body mass growth, reducing the level of the lipid peroxide, and increasing the activity of superoxide dismutase enzyme during summer [79]. Supplementation of 30 mg of Zinc (Zn) and 600 mg of Magnesium (Mg) per kg of feed improved live weight gain, feed intake, and hot and chilled dressing percentage in the heat-stressed quails [80]. The supplementation of zinc (60 mg/kg of feed) in the diet of egg-laying Japanese quail was also associated with reduced MDA concentration, increased serum vitamin C and vitamin E level, and egg production [81]. In laying hens, dietary supplementation of zinc (80–100 mg/kg of feed) [82,83] as Zn-methionine was effective in improving the eggshell thickness and mitigating the eggshell defects seen in the laying hens under heat stress.

##### Chromium

Chromium is an essential mineral, which is an integral component of chromodulin and is also necessary for insulin functioning [84]. Moreover, chromium is also involved in carbohydrate, protein, lipid, and nucleic acid metabolism [85]. Sahin et al. [86] researched the effects of chromium supplementation (chromium picolinate CrPic) at different doses (200, 400, 800 or 1200 µg/kg of feed) in heat-stressed broilers, where they found that increased supplementation of chromium was associated with an increase in body weight, feed intake, and carcass quality. They also observed a decreased level of serum corticosterone, serum glucose, cholesterol, and increased serum insulin level. Moreover, the organic form of chromium supplemented as chromium methionine was also found to improve the cellular and humoral immune responses in broilers during heat stress [87]. In laying hens, dietary supplementation of 0.4–2 mg chromium/kg of feed as CrPic improved immune response, egg quality, Haugh unit [88,89], and reduced serum glucose, cholesterol, and triglyceride concentration [90].

##### Selenium

Selenium is a vital component of at least 25 different selenoproteins, most of which are the different parts of the enzymes, such as glutathione peroxidase and thioredoxin reductases [91,92]. Type I deiodinase enzyme is one such enzyme that helps in the conversion of thyroxin into active triiodothyronine [93]. Two different forms of selenium, i.e., inorganic forms (sodium selenite and selenite) and organic forms (selenomethionine and selenium-yeast) are used as supplements for poultry. The organic forms are more easily absorbed than inorganic forms [94]. Dietary supplementation of selenium (0.3 mg/kg of feed) is found to improve the live weight and FCR in broilers during heat stress [95]. Similarly, supplementation of sodium selenite at 0.1 or 0.2 mg/kg of feed improved the carcass quality and performance of quails reared under high temperature [64]. Selenium is found to improve the productive and reproductive performance of laying hens [96] Supplementation of the selenized yeast in the diet of laying hens also improved the egg weight, egg production, Haugh units, and eggshell strength during heat stress [97]. In laying quails, there was a linear increase in feed intake, body weight, and egg production; and improvement in feed efficiency upon selenium supplementation (0.15 and 0.30 mg/kg of feed sodium selenite or selenomethionine) under heat stress [98]. They reported that Haugh units and eggshell weights were also increased upon supplementation of both organic and inorganic selenium.

##### Electrolytes

Panting in heat-stressed bird alters the acid-base balance in blood plasma and ultimately leads to respiratory alkalosis. This acid-base imbalance can be recovered by supplementation of electrolytes such as NH_4_Cl, NaHCO_3_, and KCl. During respiratory alkalosis, birds excrete a higher amount of bicarbonate ions from the kidney to restore normal blood pH. These bicarbonates ions are further coupled with Na^+^ and K^+^ ions before being excreted through the kidney. Ultimately, the loss of ions results in an acid-base imbalance [99]. Thus, sodium and potassium supplementation is preferred in heat-stressed birds to increase the blood pH and blood HCO_3_^−^, while chloride is supplemented to reduce these parameters [100]. A higher range of dietary electrolyte balance (DEB), i.e., 200–300 mEq/kg, has been suggested to be effective in ameliorating the detrimental effects of heat stress in poultry [101]. Several studies have shown sodium bicarbonate (NaHCO_3_) as the salt of choice during heat stress as it contains Na^+^ and HCO_3_^−^ [101]. Moreover, supplementation of NaHCO_3_ in heat-stressed laying hens is also found to improve eggshell quality [77]. Incorporation of NaHCO_3_ (up to 0.5%) into broiler diets also enhanced the performance of heat-stressed broiler birds [102]. Similarly, Smith et al. [103] found that dietary levels of 1.5–2.0% K from KCl were effective in improving FCR during chronic heat stress conditions. Besides including these salts in the diet, supplementation of 0.2% NH_4_Cl or 0.15% KCl, 0.6% KCl, 0.2% NaHCO3, and carbonated water in drinking water also improved the performance in the heat-stressed broiler chickens [36].

#### 3.1.6. Supplementation of Phytochemicals

Different types of phytochemicals have been supplemented in the diet to mitigate heat stress in poultry. Some of them are discussed here.

##### Lycopene

Lycopene is a predominant carotenoid mainly found in tomatoes and tomato products, and is known to enhance the production of antioxidant enzymes through activation of antioxidant response element in the DNA [104]. Supplementation of lycopene (200 or 400 mg/kg of feed) in heat-stressed broilers improved the cumulative feed intake, body weight, and FCR [105]. Lycopene is found to improve the level of antioxidant enzymes such as superoxide dismutase (SOD) and glutathione peroxidase (GSH-Px) in broilers [104]. In laying hens, dietary supplementation of lycopene improved oxidative status, enhanced vitamin levels in the egg, and also improved oxidative stability and yolk color of the egg [104].

##### Resveratrol

Resveratrol is natural bioactive polyphenols mainly found in grapes, peanuts, berries, and turmeric. Previous studies have shown that supplementation of resveratrol (400 mg/kg of feed) enhanced the antioxidant capacity in the broilers during heat stress [106]. Supplementation of resveratrol at 300 or 500 mg/kg of feed improved the average daily gain, decreased the rectal temperature, lowered the level of corticosterone, adrenocorticotropin hormone, cholesterol, and MDA in yellow-feather broilers under heat stress [107]. Additionally, in the same study, resveratrol also increased the level of triiodothyronine, glutathione, total superoxide dismutase, catalase, and glutathione peroxidase during heat stress [107]. Resveratrol also improved different gut health parameters such as microbial profile, villus-crypt structure, and expression of the tight junction and adherence junction related genes in the heat-stressed broilers [108] Interestingly, resveratrol improved meat quality in the heat-stressed broilers by increasing the muscle total antioxidant capacity (T-AOC) and activity of antioxidant enzymes (catalase, GSH-Px) [109]. In laying hens, supplementation of 200 mg resveratrol/kg of feed improved the egg production, while 400 mg resveratrol/kg of feed reduced the total serum cholesterol and triglycerides, reduced egg cholesterol content, improved antioxidant activity, and improved egg sensory scores [110].

##### Epigallocatechin Gallate (EGCG)

Epigallocatechin gallate (EGCG) is the polyphenols present in green tea extract that possess high antioxidant and anti-inflammatory properties. Luo et al. [111] used different dosages of EGCG in the feed (0, 300 and 600 mg/kg) of heat-stressed broiler birds where they found a linear increase in body weight, feed intake, and level of serum total protein, glucose and alkaline phosphatase activity in the heat-stressed birds. In a similar experiment, Xue et al. [112] reported improvement in the body weight and antioxidant enzymes (GSH-Px, SOD, and catalase) in the liver and serum of heat-stressed broiler birds with the dietary inclusion of EGCG. Sahin et al. [113] supplemented 200 or 400 mg of EGCG/kg of feed in heat-stressed female quails where they observed that increased supplementation of the EGCG linearly increased feed intake, egg production, hepatic SOD, catalase, and GSH-Px activity and resulted in a linear decrease of hepatic MDA level.

##### Curcumin

Curcumin is the primary polyphenols extracted from turmeric and possesses antioxidant and anti-inflammatory properties [114]. As animals readily absorb curcumin, its use as a potential compound to mitigate heat stress in poultry has received attention in recent years [106]. Previous studies have shown that feed with curcumin improves the growth performance of heat-stressed broiler birds [115,116,117]. Zhang et al. [115] found that the inclusion of curcumin at 100 mg/kg of feed significantly improved the final body weigh in broilers under heat stress conditions. Curcumins fortification reduced the mitochondrial MDA level; reduced the ROS production by increasing the activity of Mn-SOD, GSH-Px, Glutathione S-transferase (GSST) [117] and increased gene expression of thioredoxin-2 and peroxiredoxin-3 [115] during heat stress in broilers. In laying hens, supplementation of 150 mg/kg of feed with curcumin improved the laying performance, egg quality, antioxidant enzyme activity, and immune function during heat stress [118].

#### 3.1.7. Supplementation of Osmolytes

##### Betaine

Betaine is a small zwitterionic quaternary ammonium compound found in microorganisms, animals, and plants [119]. Betaine is incorporated in the animal diets in different forms; as anhydrous betaine, betaine monohydrate, or betaine hydrochloride [120]. Betaine possesses two fundamental metabolic activities, i.e., methyl donor activity and osmotic activity. Under heat stress, betaine plays a vital role in regulating the cellular osmotic environment, preventing dehydration by increasing the water-holding capacity of the cell [120]. Furthermore, betaine is also found to have anti-inflammatory properties and improves the intestinal function [121]. During heat stress, supplementation of betaine ranging from 0.05–0.20% improved the feed intake, carcass trait, and egg production parameters in broilers, layers, and ducks [120]. Chand et al. [122] investigated the effect of betaine in chronic heat-stressed broilers by using three different (1, 1.5, and 2 g/kg of feed) dosages of betaine. They reported significant improvement in the feed intake, weight gain, and FCR for the higher level of betaine. Furthermore, they also found a lower H/L ratio and improvement in the dressing percentage in the treatment groups supplemented with betaine. In another study, besides growth performance parameters, dietary supplementation of betaine during the cyclic heat stress condition improved digestive function and carcass traits in indigenous yellow-feathered broilers [123]. Betaine supplementation (1 g/kg of feed) also increased feed intake, protein digestibility, dressing out percentage, and improved FCR in slow-growing chicks [124]. In laying hens, supplementing betaine (1000 mg/kg of feed) along with vitamin C (200 mg/kg of feed) improved laying performance during the chronic heat stress [125]. In roosters, supplementation of betaine (1000 mg/kg of feed) improved sperm concentration and livability, seminal plasma total antioxidant capacity, fertility, and welfare under chronic heat stress [126].

##### Taurine

Taurine, 2-aminoethanesulfonic acid, is one of the most abundant amino acids distributed in different parts of animal tissues [127]. Taurine plays a role in antioxidant action, bile acid conjugation, maintenance of calcium homeostasis, osmoregulation, and membrane stabilization [127]. The use of taurine to mitigate heat stress in poultry has gained popularity in recent days under chronic heat stress. Supplementation of 0.1% taurine in the drinking water demonstrated significant improvement in the final body weight of chronic heat-stressed broilers. Moreover, expression of heat shock proteins was lowered in the taurine supplemented broilers indicating improved thermotolerance in these birds under heat stress [128]. Similarly, He et al. [129] reported that supplementation of taurine (5 g/kg of feed) in broilers under heat stress improved jejunal morphology, decreased the concentrations of serum ghrelin, increased the concentrations of somatostatin and peptide YY in the duodenum and increased the expression of appetite-related genes [129]. Taurine supplementation was found to reduce fat deposition in the liver of chronic heat-stressed broilers [130]. Supplementation of the taurine (0.1% of feed) in the laying hen exhibited enhanced oviductal health and reduced oviductal injury [131]. Taurine supplementation in the laying hens under heat stress, however, is not well studied, and thus, further research is warranted.

**Table 1 animals-10-01266-t001:** Summary of the beneficial effects of vitamins, minerals, phytochemicals, and osmolytes in heat-stressed poultry.

Supplements	Beneficial Effects on Heat-Stressed Birds	References
Vitamin E	➢ prevent liver damage, facilitate the synthesis and release of vitellogenin; ↑ egg production in laying hen	[62]
	➢ ↓ liver and serum MDA concentration; ↑ increased serum and liver vitamin E and A concentration in broilers	[64]
Vitamin A	➢ ↑ egg weight in laying hens	[68]
	➢ ↑ live weight gain, improved feed efficiency and ↓ serum MDA concentration in broilers	[69]
Vitamin C	➢ improved growth rate, nutrient utilization, egg production, and quality, immune response, and antioxidant status in poultry	[72]
	➢ ↓ serum concentration of MDA, homocysteine, and adrenal corticotrophin hormone in Japanese quail	[73]
	➢ improved body weight gain and FCR in broilers	[74]
Zinc	➢ improved body mass growth, ↓ level of the lipid peroxide, ↑ activity of SOD in broilers	[79]
	➢ improved live weight gain, feed intake, and hot and chilled dressing percentage in quails	[80]
	➢ ↓ MDA concentration, ↑ serum vitamin C and vitamin E level, ↑ egg production in Japanese quail	[81]
	➢ improved eggshell thickness and mitigate the eggshell defects in laying hens	[82,83]
Chromium	➢ ↑ body weight, feed intake, and carcass quality; ↓ level of serum corticosterone concentration; ↓ serum glucose and cholesterol concentration; ↑ serum insulin level in broilers.	[86]
	➢ improved cellular and humoral immune responses in broilers	[87]
	➢ ↑ immune response, egg quality, Haugh unit	[88,89]
	➢ ↓ serum glucose, cholesterol, and triglyceride concentration	[90]
Selenium	➢ improved live weight and FCR	[95]
	➢ improved egg production, egg weight, Haugh unit and eggshell strength in laying hens	[97]
	➢ increased of feed intake, body weight and egg production in quails	[98]
Sodium Bicarbonate	➢ improved eggshell quality in laying hens	[77]
KCL	➢ improved FCR in broilers	[103]
Lycopene	➢ ↑ cumulative feed intake and body weight; ↓ FCR in broilers	[105]
	➢ ↑ antioxidant level enzymes (SOD, GSH-Px) and ↓ MDA concentration in broilers	[104]
	➢ ↑ oxidative status of laying hens, enhanced vitamin levels in the egg; improved egg oxidative stability and yolk color	[104]
Resveratrol	➢ ↑ average daily gain, ↓ rectal temperature, ↓ corticosterone, adrenocorticotropin hormone, cholesterol, and malonaldehyde; ↑ triiodothyronine, glutathione, total superoxide dismutase, catalase, and glutathione peroxidase in yellow-feather broilers	[107]
	➢ improved microbial profile, villus-crypt structure, and expression of the tight junction related genes in broilers	[108]
	➢ ↑ muscle T-AOC and activity of antioxidant enzymes (catalase, GSH-Px)	[109]
	➢ ↓ total serum cholesterol and triglycerides, ↓ egg cholesterol content, ↑ antioxidant activity, and ↑ egg sensory scores	[110]
Epigallocatechin gallate (EGCG)	➢ ↑ body weight, feed intake, and level of serum total protein, glucose, and alkaline phosphatase activity in broilers	[111]
	➢ improved in level of antioxidant enzymes (GSH-Px, SOD, and catalase) in the liver and serum in broilers	[112]
	➢ ↑ feed intake, egg production, hepatic SOD, catalase, and GSH-Px activity; ↓ hepatic MDA level in quails	[113]
Curcumin	➢ ↓ mitochondrial MDA level; ↑ activity of Mn-SOD, GSH-Px, GSST in broilers	[117]
	➢ ↑ gene expression of thioredoxin 2 and peroxiredoxin-3 in broilers ➢ improved the laying performance, egg quality, antioxidant enzyme activity, and immune function during heat stress in laying hens	[115,118]
Betaine	➢ improvement in the feed intake, weight gain, and FCR; lower H/L ratio; improvement in the dressing percentage in broilers	[122]
	➢ improved digestive function and carcass traits in indigenous yellow-feathered broilers	[123]
Taurine	➢ improved expression of heat shock proteins and body weight in broilers	[128]
	➢ improved jejunal morphology, ↓ concentrations of serum ghrelin, ↑concentrations of somatostatin and peptide YY in the duodenum; ↑ expression of appetite-related genes	[129]

### 3.2. Genetic Approach

Improved broiler lines have a higher metabolic rate; as a result, they are more susceptible to heat stress. Thus, developing poultry lines incorporating some of the genes that help to reduce heat stress can be instrumental in further excelling the production traits of these breeds in the hot and arid areas.

#### 3.2.1. Naked Neck (Na) Gene

Na gene is the single dominant autosomal gene that helps to reduce feathers in the neck region, thus helps to dissipate heat through the neck region in birds. The naked neck gene reduces the feather cover by 20% and 40% in Na/na (heterozygous necked neck) and Na/Na (homozygous necked neck), respectively, as compared to normal siblings (na/na) [132]. Na gene in broilers is associated with the increase in breast muscle and body weight [133,134], reduce abdominal fat [135], and body temperature [136]. The total plasma cholesterol level and H/L ratio were significantly lowered in the naked necked birds as compared to typical birds during the summer season [137]. Laying birds with a naked neck gene also displayed an improvement in egg mass, number, and quality under hot temperatures [138]. These studies indicate that there is a scope of incorporating such genes to develop a chicken breed that can cope with heat stress. 

#### 3.2.2. Frizzle Gene

The frizzle (F) gene causes the curving of the outline of the feather resulting in a reduced featherweight and insulating property of the feather cover and increases heat radiation from the body [36]. Homozygous frizzle gene in laying hens improved the egg production and quality traits by increasing the magnitude of heat dissipation as compared to heterozygous carriers and normal feathered hens [139]. Sharifi et al. [140] reported a significant interaction between feathering genotype (FF) and environmental temperature for all reproductive traits (egg production, hatchability, and chick production) except sexual maturity under heat stress. At higher temperatures, they reported a distinct reduction in all reproductive traits except sexual maturity for normally feathered hens compared with frizzle-feathered hens, whereas under lower temperatures (19 °C), egg production, and the number of chicks of the FF genotype were reduced and sexual maturity was delayed. 

The beneficial effect of the F gene as compared to the Na gene is lower in broilers at high temperatures. However, there is an additive effect in the double heterozygous (Na/Na F/f) broiler [141]. So, the frizzle gene is another potential target for developing heat-tolerant chickens. 

#### 3.2.3. Dwarf (dw) Gene

The dwarf gene is a sex-linked recessive gene associated with reduced body weight by about 40% and 30% in homozygous males and females, respectively [54]. There has been a discrepancy regarding the advantage of the dw gene in the heat-stressed laying hens. Decuypere et al. [142] concluded that the inherent heat tolerance of dw genotype in laying hens was uncertain. It has been found that the dw gene in fast-growing broiler chickens under chronic heat stress conditions did not improve heat tolerance [143].

### 3.3. Housing

Naturally ventilated open-type housing is most common in the tropics, which should be oriented in the east-west direction [144]. The width of such housing should not exceed 12 m, while the length of the building can depend upon the convenience. In the case of long buildings, doors should be placed at an interval of 15–30 m. It is recommended to have a side-wall height of at least 2.1 m along with curtails that can be raised or lowered easily. Regarding the roof, a roof slope of 45 °C is recommended as it reduces the heat gain of the roof from direct solar radiation [54]. It has been observed that farmers used different local materials such as thatched and bamboo to insulate the roof. In the case of an uninsulated metal roof, a sprinkling roof with cool water has also been a common practice to reduce heat load in poultry houses [54]. Moreover, in this kind of housing, fans (either suspended from the interior building structures or vertical ceiling fans), interior fogging, and sprinkling systems have been used effectively [54].

With the advancement of technologies, there has been a surge in the use of a closed house system for more intensive farming systems recently [145]. Closed housed systems equipped with air conditioning, cooling pads, cool perches, and exhaust fans are found useful in attenuating the negative effects of heat stress in poultry. However, such houses are expensive to build and operate in developing nations [146], and therefore dietary manipulations are more appropriate.

### 3.4. Others

In addition to the aforementioned strategies, some other strategies have been used to combat heat stress in poultry, such as early heat conditioning (EHC) [36], early feed restriction (EFR) [36], reducing stocking density of birds [8,147], and thinning the litter during summer seasons. In EHC, birds are exposed to high temperatures (36 °C) for 24 h at 3 to 5 d of age [36], while in EFR about 60% of feed is restricted on days 4, 5, and 6 [36]. EHC and EFR developed the tolerance capacity of birds against high temperature during the later growth stage before marketing [36]. EHC may play a role in the acquisition of heat tolerance capacity by suppressing the expression of an uncoupled protein (avUCP) [148] and by improving the expression of HSP70 [149] while EFR might possess beneficial effects in heat stress by improving the expression of HSP70 [149]. Reducing the stocking density of birds increases the feed and water accessibility, and also increases heat dissipation from the body [8]. Thinning of the litter helps to make the litter dry, making it favorable for birds to cool their body by dust bathing.

In ovo supplementation of nutrients is known to induce post-hatch immunity, antioxidant indices, and growth performances. Sulfur amino acids are known to play a crucial role in protein structure, metabolism, immunity, and oxidation [150]. Recently, in ovo inoculation of sulfur-containing amino acids in heat-stressed embryo induced serum antioxidant indices and antioxidant related genes expression, reduced HSP70 gene expression, corticosterone concentrations, and lipid profile in hatched broiler chicks [151]. Dietary supplementation of N-acetylcysteine improved the growth performance and intestinal function of broilers exposed to heat stress [152]. N-acetylcysteine also mitigated heat stress in breeder Japanese quail under heat-stressed conditions [153]. Thus, further studies are required to delineate the dietary supplementation of Sulfur amino acids in heat-stressed broiler chickens and laying hens.

## 4. Conclusions

With the rising global temperature, heat stress has been a severe challenge to the growth of the poultry industry. Several strategies have been tried and tested to counteract heat stress in poultry. However, only a few of them are widely used in the poultry industry. Heat stress in poultry results from the interplay of several factors, such as high environmental temperature, humidity, radiant heat, and airspeed, and causes several physiological, neuroendocrine, and behavioral changes. So, no single approach alone is enough to negate the impacts of heat-stress on poultry. Therefore, there is a need for a holistic approach to attenuate the negative effect of heat stress in poultry. The potential use of Na and F genes, along with proper nutrition, housing, and management should be beneficial in mitigating heat stress. Further research testing a combination of some approaches for ameliorating heat-stress mentioned in this article, to observe their efficiency and cost-benefit in the poultry industry is warranted.

## Figures and Tables

**Figure 1 animals-10-01266-f001:**
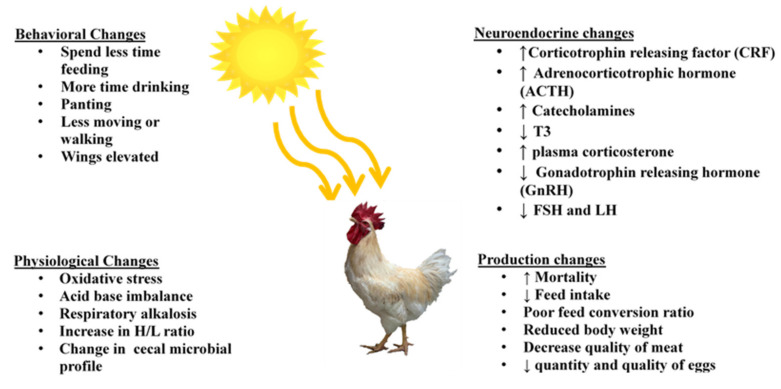
Effects of heat stress on behavioral, physiological, neuroendocrine, and production traits.

**Figure 2 animals-10-01266-f002:**
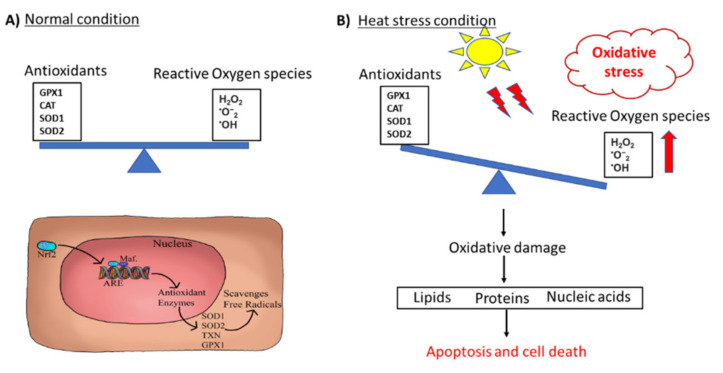
Schematic diagram showing the redox system. (**A**) Normal condition, and (**B**) under heat stress.

**Figure 3 animals-10-01266-f003:**
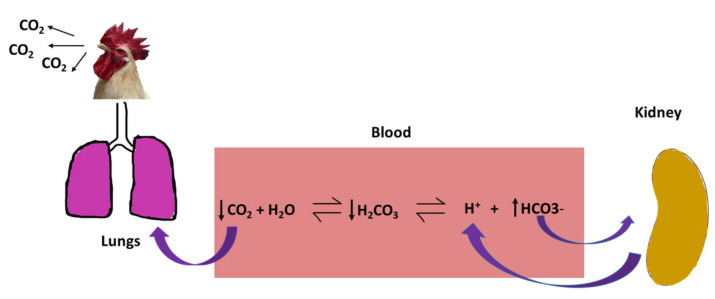
Schematic diagram showing an acid-base imbalance in poultry under heat stress.

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
