# Peer review of "Impact of Heat Stress on Poultry Health and Performances, and Potential Mitigation Strategies"

_animals, 2020, doi:10.3390/ani10081266_

Round 1

Reviewer 1 Report

General comments:

The manuscript discusses heat stress as an important area of poultry production. The manuscript is well written and provides a comprehensive summary of effects of and intervention strategies to negate the effects of heat stress in commercial poultry.

Minor comments:

Ln 10: delete “as consequence of global warming’ and add ‘by causing heat stress’ at the end of the sentence

Ln 11: isn’t ‘neuroendocrine’ a part of physiology

Ln 15: replace ‘most challenging’ with ‘major’

Ln 16: rephrase as ‘Heat stress causes several physiological effects…..’

Ln 23: add ‘certain’ before ‘breed lines have also gained…..’

Ln 42: 150 eggs in early 90’s is incorrect.. please correct with proper timeline

Ln 49: replace ‘be way higher in recent years’ with ‘increase in coming years’

Author Response

General comments:

The manuscript discusses heat stress as an important area of poultry production. The manuscript is well written and provides a comprehensive summary of effects of and intervention strategies to negate the effects of heat stress in commercial poultry.

Response: We highly appreciate the reviewers’ thorough reading and insightful comments and suggestions on our manuscript. We have responded to the reviewers’ comments and have revised the manuscript accordingly in the text.

Ln 10: delete “as consequence of global warming’ and add ‘by causing heat stress’ at the end of the sentence

Response: We really appreciate the reviewer’s suggestion for the improvement of the manuscript. We have now made a change in our revised manuscript. (current line 10)

Ln 11: isn’t ‘neuroendocrine’ a part of physiology

Response: We have replaced ‘neuroendocrine’ with ‘production’. (current line 11)

Ln 15: replace ‘most challenging’ with ‘major’

Response: We have replaced the ‘most challenging’ with ‘major’ in our revised manuscript. (current line 14)

Ln 16: rephrase as ‘Heat stress causes several physiological effects….’

Response: We have as rephrase as ‘Heat stress causes several physiological changes….’ in our revised manuscript. (current line 15)

Ln 23: add ‘certain’ before ‘breed lines have also gained….’

Response: We have now added ‘certain’ in our text. (current line 22)

Ln 42: 150 eggs in early 90’s is incorrect. please correct with a proper timeline

Response: Laying hens in the early 1900s used to lay 150 eggs per year while current commercial laying hens lay around 300 eggs annually. We sourced this info from American Egg Board:  https://www.aeb.org/farmers-and-marketers/history-of-egg-production

Ln 49: replace ‘be way higher in recent years’ with ‘increase in coming years’

Response: We have made changes as per the reviewer’s suggestion. (current line 47)

Reviewer 2 Report

This is a good review paper that updated all aspects of heat stress and ways of alleviating it. Heat stress is a worldwide problem and hot topic and will remain forever, due to global warming and the rising earth temperature. This has a severe challenge to animal production worldwide. Strategies have been developed and examined tested to compact heat stress.  The potential of this paper is very clear and fits the journal readers and thus deserves publication.  The paper is well written, discussed, and presented and updated the current knowledge in compacting heat stress.

All my comments are shown in the attached copy.

Author Response

This is a good review paper that updated all aspects of heat stress and ways of alleviating it. Heat stress is a worldwide problem and hot topic and will remain forever, due to global warming and the rising earth temperature. This has a severe challenge to animal production worldwide. Strategies have been developed and examined tested to compact heat stress.  The potential of this paper is very clear and fits the journal readers and thus deserves publication.  The paper is well written, discussed, and presented and updated the current knowledge in compacting heat stress.

All my comments are shown in the attached copy.

Response: We highly appreciate the reviewers’ thorough reading and insightful comments and suggestions on our manuscript. We have responded to the reviewers’ comments and have revised the manuscript accordingly in the text, and the response is also provided here as per lines.

Line 42:

Response: Thank you for your suggestion. This was the information that we used from Zuidhof et al. 2014. The weight mentioned was that of the 1950s and 2005 in 56 days. We have now clearly mentioned this information in revised manuscript. (current line 40)

Line 155:

Response: We do not think that there is any defined period for restricting the feed during heat stress. But at the same time, we too agree with the reviewer’s suggestion to mention (8 am to 5 pm) as it is the hottest part of the day. We have mentioned this information in our revised manuscripts. (current line 154)

Line 163

Response: It is now mentioned in line 157.

Line 184 Fat

Response: According to the reviewer’s suggestions, we have now added references in the text (current line 189-193)

Attia, Y.A.; Al-Harthi, M.A.; Sh Elnaggar, A. Productive, physiological and immunological responses of two broiler strains fed different dietary regimens and exposed to heat stress. Ital. J. Anim. Sci. 2018, 17, 686–697, doi:10.1080/1828051X.2017.1416961.

Attia, Y.A.; Hassan, S.S. Broiler tolerance to heat stress at various dietary protein/energy levels. Eur. Poult. Sci. 2017, 81, doi:10.1399/eps.2017.171.

Line 196 Vitamin E

Response: According to the reviewer’s suggestions, we have now added references in the text (current line 211-213).

Attia, Y.A.; Al-Harthi, M.A.; El-Shafey, A.S.; Rehab, Y.A.; Kim, W.K. Enhancing tolerance of broiler chickens to heat stress by supplementation with Vitamin E, Vitamin C and/or probiotics. Ann. Anim. Sci. 2017, 17, 1155–1169, doi:10.1515/aoas-2017-0012.

Line 265 Selenium

Response: We have now added information and updated references. (current line 277)

Attia, Y.A.; Abdalah, A.A.; Zeweil, H.S.; Bovera, F.; Tag El-Din, A.A.; Araft, M.A. Effect of inorganic or organic selenium supplementation on productive performance, egg quality and some physiological traits of dual-purpose breeding hens. Czech J. Anim. Sci. 2010, 55, 505-519.

Line 338

Response: We have updated the references. (current line 343)

Attia, Y.A.; Al-Harthi, M.A.; Hassan, S.S. Turmeric (Curcuma longa Linn.) as a phytogenic growth promoter alternative for antibiotic and comparable to mannan oligosaccharides for broiler chicks. Rev. Mex. Ciencias Pecu. 2017, 8, 11–21, doi:10.22319/rmcp.v8i1.4309.

Line 351

Response: We have now updated the information and references. (current line 369-375)

Attia, Y.A.; Hassan, R.A.; Qota, E.M.A. Recovery from adverse effects of heat stress on slow-growing chicks in the tropics 1: Effect of ascorbic acid and different levels of betaine. Trop. Anim. Health Prod. 2009, 41, 807–818, doi:10.1007/s11250-008-9256-9.

Attia, Y.A.; El-Naggar, A.S.; Abou-Shehema, B.M.; Abdella, A.A. Effect of supplementation with trimethylglycine (Betaine) and/or vitamins on semen quality, fertility, antioxidant status, dna repair and welfare of roosters exposed to chronic heat stress. Animals 2019, 9, 547, doi:10.3390/ani9080547.

Line 437

Response: We have now added ‘and thus dietary manipulation are more appropriate’ as suggested by the reviewer.

Line 454 please extend the challenges to the application and your view to solve to overcome such challenges

Response: Most of the studies in heat stress are done in a controlled environment mostly just considering the temperature and humidity. In the application at the farm level, however, different factors such as radiant heat, airspeed, etc. exacerbate heat stress conditions in poultry. Thus, supplementation of just vitamins or minerals or osmolytes maybe not be effective to attenuate the negative effects of heat stress in poultry. Therefore, we have to take a holistic approach considering proper nutrition, management, housing, and improved breeds with heat-tolerant genes. These things are mentioned in the revised manuscript (current line 461-467).

In our opinion, genetic improvement of the breed along with nutritional reprogramming will help in reducing heat stress effects in poultry.

Reviewer 3 Report

This manuscript is a well summarized and organized review of the heat stress as an climate change issue.  Reviews also includes the potential mitigation strategies.  There is only a few minor suggestions.

  1. The title of this article suggests the main purpose is to discuss the impact of heat stress on "poultry health and performances",  however the authors actually only focus on broiler chickens and laying hens.  The poultry industry should include turkey and water fowls (ducks, geese).
  2. For the supplements in the diet, in 2.2 supplementation of vitamins, minerals, and eletrolytes, both Vitamin E and Selenium were included, yet the S-containing amino acids: cysteine was not mentioned.  Normally, these three nutrients are tightly bound together and should be all included.

Author Response

This manuscript is a well summarized and organized review of the heat stress as an climate change issue.  Reviews also includes the potential mitigation strategies.  There is only a few minor suggestions.

Response: We highly appreciate the reviewers’ thorough reading and insightful comments and suggestions on our manuscript. We have responded to the reviewers’ comments and have revised the manuscript accordingly in the text.

  1. The title of this article suggests the main purpose is to discuss the impact of heat stress on "poultry health and performances",  however, the authors actually only focus on broiler chickens and laying hens.  The poultry industry should include turkey and waterfowls (ducks, geese).

Response: We totally agree with the reviewer’s suggestion about incorporating other commercial birds. However, it is difficult to include all the poultry species in one review paper so we focused on the predominantly reared poultry species (broiler chickens and laying hens) and described in our paper. In the future, we are aiming to summarize the findings of the Turkeys and waterfowls individually.

  1. For the supplements in the diet, in 2.2 supplementation of vitamins, minerals, and electrolytes, both Vitamin E and Selenium were included, yet the S-containing amino acids: cysteine was not mentioned.  Normally, these three nutrients are tightly bound together and should be all included.

Response: We really appreciate the reviewer’s suggestion on this. In this review paper, we have included all the supplements that are being used in the poultry industry under heat stress. We found the extensive number of studies on Vitamin E and Selenium as a supplement in heat-stressed birds. However, there are a very limited studies on cysteine as a supplement to mitigate heat stress in poultry. Based on the reviewer’s suggestions, we have included some of the findings from in ovo inoculation of Sulfur AA, in broiler chicken and Japanese Quail (Line 469-478).